# The Impact of Different Parameters on the Formwork Pressure Exerted by Self-Compacting Concrete

**DOI:** 10.3390/ma16020759

**Published:** 2023-01-12

**Authors:** Yaser Gamil, Andrzej Cwirzen, Jonny Nilimaa, Mats Emborg

**Affiliations:** Building Materials, Department of Civil, Environmental and Natural Resources Engineering, Lulea University of Technology, 97187 Luleå, Sweden

**Keywords:** formwork, pressure, self-compacting concrete, parameters

## Abstract

Despite the advantageous benefits offered by self-compacting concrete, its uses are still limited due to the high pressure exerted on the formwork. Different parameters, such as those related to concrete mix design, the properties of newly poured concrete, and placement method, have an impact on form pressure. The question remains unanswered on the degree of the impact for each parameter. Therefore, this study aims to study the level of impact of these parameters, including slump flow, T500 time, fresh concrete density, air content, static yield stress, concrete setting time, and concrete temperature. To mimic the casting scenario, 2 m columns were cast at various casting rates and a laboratory setup was developed. A pressure system that can wirelessly and continuously record pressure was used to monitor the pressure. Each parameter’s impact on the level of pressure was examined separately. Casting rate and slump flow were shown to have a greater influence on pressure. The results also demonstrated that, while higher thixotropy causes form pressure to rapidly decrease, a high casting rate and high slump flow lead to high pressure. This study suggests that more thorough analysis should be conducted of additional factors that may have an impact, such as the placement method, which was not included in this publication.

## 1. Introduction

The formwork pressure exerted by self-compacting concrete (SCC) is considerably higher than that of normal concrete due to its high flowability ([1]). This has slowed down the acceptance of SCC in civil engineering projects [2]. The current design models assume full hydrostatic pressure while designing the form ([1,3,4]). International standards such as ACI347R [5] and the European guidelines [6] have also suggested the design to be fully hydrostatic. However, other findings have proven otherwise, showing that the actual form pressure is less than the hydrostatic pressure [7,8,9,10,11]. This raises the question as to how much lower the actual pressure can be when casting with SCC. This also introduces the possibility of using SCC more often and shifting from a reliance on normal concrete.

Previous studies have shown that form pressure is affected by several parameters, and these parameters are generally classified into parameters related to mixture design and material properties, the fresh concrete properties of the mix, and the placement method and speed of casting [12]. However, the degree of impact of these parameters is still unknown.

The parameters that affect the amount of pressure and its distribution over the form’s height are relatively interrelated. For example, mixture design affects the flowability of the concrete and has an influence on the pressure and its reduction over time. To understand that phenomenon, [13] conducted a study incorporating factorial design to investigate the impacting parameters of fresh normal concrete on pressure. Their analysis showed that formwork shape and coarse aggregate have a minor impact on pressure, temperature has an inverse relationship with pressure, and form size has a major effect on pressure, with narrow sections producing less pressure. Similarly, [14,15] found that wall friction forces between concrete and formwork affect the amount of pressure, with higher friction leading to higher pressure.

A different study by [16] investigated the effect of water-to-cement ratio and the amount of water-reducing admixture and found that both thixotropy and form pressure are affected by the w/cm ratio, whereby mixes with 0.46 w/cm showed greater initial pressure compared to mixes with 0.4 and 0.36. The reason for this is thought to be the increased water paste content and a reduction in coarse aggregate, which thus lowers the shear strength of fresh concrete. Additionally, [17] studied the effect of viscosity-enhancing admixtures (VEA) on the pressure of SCC and found that the use of VEA results in lower form pressure, while water-reducing admixtures exhibited higher pressure with higher dosages.

The important effect of reinforcing bars has also been demonstrated, as shown in congested reinforcement where pressure was rather lower than it was when few reinforcing bars were used [8,18,19]. The effect of mixture composition has also been shown to have high importance in terms of form pressure [20]. For example, powdered materials in the mix considerably affect the amount of pressure, with higher content increasing pressure and reducing workability [21].

Casting rate and placement methods have a significant effect on pressure, with a higher casting rate leading to higher pressure [1,4,16,22,23,24,25,26,27,28]. Similarly, casting characteristics also affect pressure, with concrete temperature accelerating pressure decay and greater casting depth leading to higher pressure [20,29]. In this article, an extended investigation is performed while varying slump flow, T500 time, fresh concrete density, air content in concrete, static yield stress, and the ambient and concrete temperature of concrete. Additionally, their impact on the amount of form pressure and its distribution was also investigated. The use of a state-of-the-art pressure system helped to realize more accurate and real-time pressure monitoring during the casting process, which thus allowed for instant decision making regarding the amount of pressure and casting rate.

## 2. Methodology

### 2.1. Experimental Setup

A laboratory setup was developed to simulate the casting process. A 2 m plastic pipe was vertically fixed to the column and given rigid support to avoid any tilt, trembling, or collapse (Figure 1). Pressure sensors were installed and mounted to the pipe by drilling 50 mm holes to enable direct contact with the concrete. Four sensors were installed. The first sensor was placed about 0.05 m from the ground, the second sensor 0.5 m from the ground, the third sensor 1 m from the ground, and the fourth sensor 1.5 m from the ground. The concrete was prepared and mixed in the laboratory, and the number of batches was governed by the casting rate (for example, casting 1 m/h required 2 batches of concrete). The casting was performed from the top and a plastic cone was used to avoid any spillage of concrete.

### 2.2. The Instrument for Measuring Pressure

With the help of cutting-edge sensors and the Internet of Things, a system developed by PERI was used in the experiments with the idea of turning cast-in-place concrete into a digital process. The sensor system consisted of data transmitters that provided readings from the sensors to the cloud. As seen in Figure 2, the pressure sensors were mounted to the pipe to facilitate direct contact with the concrete. Real-time data regarding pressure during casting were displayed instantly on the computer. This made it simple to check the pressure. If pressure was low, casting pace could be increased; if pressure was high, casting could be slowed down.

### 2.3. Mix Design and Test Plan

The mixture used for casting was a commercial mixture with fixed proportions, and only the type of cement and superplasticizers were altered depending on the required slump flow. Different variations were then used to study the significance level of different parameters, such as slump flow, T500 time, fresh concrete density, air content, static yield stress, concrete setting time, and concrete temperature.

For all batches, the water-to-cement ratio was kept at 0.59, and the superplasticizer (SP) used was MasterGlenium 592. A small mixer was used in the laboratory to mix the concrete in various batches. New batches of concrete were created for each layer of casting while preserving the same characteristics. To ensure the same flowability, the measurement of the slump was kept track of and maintained for each batch. Table 1 shows the initial mix design used for casting the pipes. The test plan used for varying the input parameters is described in Table 2. The test plan and mix proportions are part of a pre-established plan that was used by the authors in a different study to assess mathematical models for forecasting form pressure when casting with SCC.

### 2.4. Cement Types

Two cement types were used. The first cement was Bascement (BAS) CEM II/A-V 52.5 N Portland-fly ash, which is recommended for use in normal concrete work, per the supplier Cementa. This cement has a different setting time. The second cement, known as Anläggningscement FA (ANLFA), is a Portland-fly ash cement type (CEM II/A-V 42.5 N MH/LA/NSR) used in mass concrete which has mild heat requirements in terms of hydration (cementa.se).

### 2.5. Casting Rate

Casting rate is a significant parameter that affects the amount of lateral pressure when casting with SCC. In the laboratory plan, casting rate was set at 0.25 m/h, 0.5 m/h, 1 m/h, and 4 m/h. Using these diverse rates, the effect of casting rate was then investigated. 

### 2.6. Slump Flow

Different types of slump flow were created by controlling the amount of superplasticizers so as to produce the desired slump flow diameter, with the first interval being 700–750 mm, the second 600–650 mm, the third 500–550 mm, and the fourth interval 400–450 mm.

## 3. Testing of Fresh Concrete

Before casting, the concrete was tested for different properties, including slump flow, T500 time, air content, fresh density, setting time, and static yield stress. The tests were performed according to the guidelines specified for each test in the European guidelines for SCC [6], except for setting time, which was performed according to the German standard [25], and static yield stress, which was performed according to the portable vane test developed by [1].

The setting bag test was conducted as specified by [25] using a polyethylene plastic bag containing around 8 L of concrete in a bucket. The consistency was checked every 30 min by manually applying a force of around 50 N to the bag surface via thumb press and examining the concrete imprint. Setting time was indicated by the depth of the impression. Setting time is calculated as 1.25 times the amount of time needed to dent something less than 1 mm. For instance, if recording of setting time begins at te and kb = 5 h, the final setting time is te = 1.25 × 5 = 6.25 h.

For each concrete recipe, the portable vane test or torque test was performed instantly to measure static yield stress. This was used to investigate the structural behavior of SCC at rest and involves using four-blade vanes and a torque gauge to measure the torque required to break the structure. The static yield stress (in Pa) of the concrete at rest can be determined from the measured torque value and vane geometry. The method is explained in detail in [1]. 

## 4. Results

### 4.1. Form Pressure Monitoring

An example of pressure monitoring is shown below for one of the tests of SCC with a casting rate of 0.25 m/h. Pressure was monitored over 8 h of casting, as demonstrated in Figure 3.

Figure 3 demonstrates pressure monitoring over an 8 h period of casting on a real-time basis, with the system capable of sending a reading every minute. From the results, it can be seen that the actual pressure exerted by the concrete is far less than the hydrostatic design which the form is supposed to bear, thus suggesting that it is not a cost-effective design and that there is a possibility of casting at a rate faster than 0.25 m/h. Hydrostatic pressure was calculated using the principle formula pressure P_hydro_ = ρgh, where ρ is concrete density in kg/m^3^, h is casting height (m), and g is gravitational acceleration (9.8 m/s^2^).

Figure 4 shows selected values of the pressure across the height of the form. It is indicated that pressure is affected by casting rate, with a higher casting rate resulting in higher form pressure. This was also proven by [10], who indicated that pressure is highly affected by casting rate. By varying the amount of superplasticizer supplied, a modification in slump flow was also achieved, and the results are shown in Figure 5.

Figure 5 illustrates how higher form pressure is affected by higher initial slump flow. It is indicated that when the initial slump flow is high, the pressure remains high even if the casting level is near the top of the form. Similar findings were observed by [26]. This is explained by the pressure decay phenomenon, which is influenced by the thixotropy (or structural development) of the concrete over time; concrete that dries out more quickly starts to exert less pressure at the bottom cast layers. Observing how the type of cement affects the form is equally important, as various studies have shown [3,10,17,20]. 

The impact of employing various types of cement on pressure is demonstrated in Figure 6 through a comparison of BAS and ANLFA cement using the data collected from the bottom sensor (0.25 m/h casting rate). The fresh density of concrete produced with various types of cement varies, measuring 2331 kg/m^3^ for BAS concrete and 2382 kg/m^3^ for ANLFA concrete. The suppliers of the concrete claim that BAS takes 150 min to set, whereas ANLFA takes 170 min. Figure 6 indicates that cement hardens more quickly, with BAS showing a value of 12.0 kPa and ANLFA a value of 13.14 kPa at the bottom of the form. According to the findings, cement type has a small impact on maximum pressure, though a larger impact is seen when examining pressure decay.

Figure 7 shows the effect of different concrete densities on pressure with a fixed casting rate for both. It is indicated that higher concrete density results in higher form pressure.

Figure 8 shows the effect of different air content on form pressure while maintaining the same casting rate and cement type, with the only change being the percentage of air in the concrete. It is indicated that higher air content leads to initial low form pressure in early fresh concrete, and the effect is diminished when the concrete starts to harden. 

Figure 9 shows the effect of different ambient temperatures on the amount of form pressure. It is indicated that it is rather difficult to observe the effect of concrete temperature on pressure.

Concrete temperature affects the amount of form pressure reduced but not maximum pressure (Figure 10), with concrete at higher temperatures tending to harden faster than at lower temperatures due to the hydration rate of formerly cast concrete.

Setting time was documented according to DIN18218 [25], which is formally known as the setting bag test. The results in Figure 11 indicate that higher setting time leads to higher pressure, which is due to the rate of hardening (concrete that hardens faster leads to lower form pressure).

The effect of static yield stress at 15 min is crucial to understanding the effect of concrete viscosity as well, with high-viscosity concrete leading to lower form pressure. From Figure 12, it is indicated that using concrete with higher stress at 15 min results in lower pressure.

### 4.2. Correlation between Variables

Correlation denotes the form of relationship between variables, which in this case includes either inputs such as density, height, and casting rate or the output, which is pressure. A negative coefficient denotes that when one variable decreases the other increases, while a positive coefficient denotes that when a variable increases the other associated variable also increases.

The results in Table 3 indicate how important the variables are in terms of their impact on the parameters related to form pressure, not only through their direct importance, but also in a cumulative sense, which was obtained by analyzing all the time series data obtained during the laboratory tests. These data were sourced from random analysis in order to estimate the correlation between the input variables stated in Table 3. Values indicate the degree of relationship between the variables. The correlation falls between −1 and +1. The focus of this study is the relationship between formwork pressure and the input parameters (concrete properties). It is indicated that the parameters of density and air temperature have a different correlation with maximum pressure. Fresh concrete density, as a main parameter in hydrostatic pressure, has been identified in previous studies as a parameter that significantly affects form pressure. This was specifically addressed by [7,12]. Air temperature was also identified by [1] as a parameter that significantly affects form pressure over time.

## 5. Conclusions

This study explored the degree to which slump flow, T500 time, fresh concrete density, air content, static yield stress, concrete setting time, and concrete temperature have an impact on form pressure when casting with SCC. Extensive laboratory experiments were conducted with a newly developed pressure system to monitor pressure and study the effects of these parameters. Several conclusions are made:Casting rate has a significant effect on the amount of pressure, with faster rates resulting in more pressure.Flowability, as measured by slump flow, also has a significant effect on pressure.Cement type (ANLFA and BAS) had a minimal effect on form pressure during casting, though a larger impact was observed when examining pressure reduction.Higher density results in higher pressure.Higher air content causes early fresh concrete to have lower initial form pressure, and this impact lessens as the concrete begins to solidify.It can be challenging to determine how concrete temperature affects pressure.The temperature of the concrete affects the amount of form pressure that can be reduced but does not affect maximum pressure; higher temperature concrete tends to harden more quickly than lower temperature concrete due to the hydration rate of previously cast concrete.Concrete that sets more slowly leads to higher pressure, whereas concrete that sets more quickly leads to lower form pressure.Concrete with high viscosity has less form pressure, which is supported by evidence that, after 15 min, pressure is lower in concrete with increased stress.The correlation matrix demonstrated that air temperature and density both have a greater correlation with maximum pressure, as shown by measurements recorded throughout the entire testing period.

## Figures and Tables

**Figure 1 materials-16-00759-f001:**
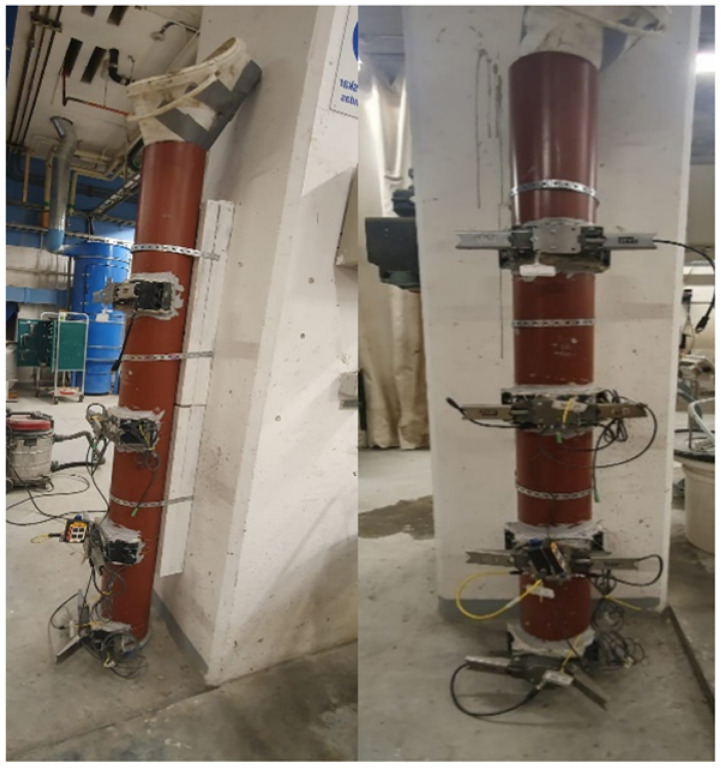
Laboratory setup.

**Figure 2 materials-16-00759-f002:**
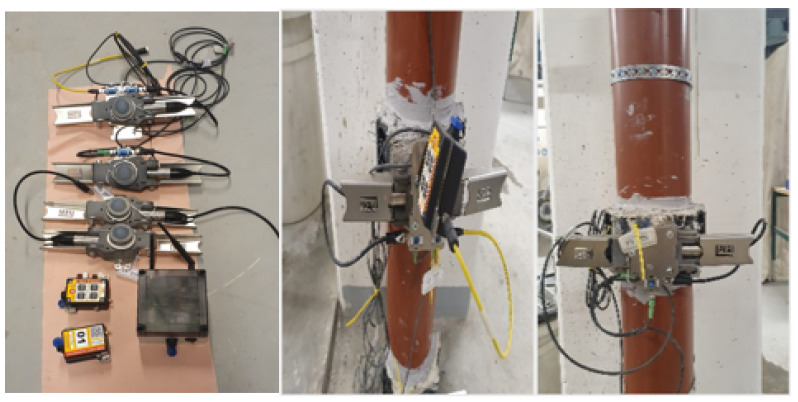
Pressure measuring system (the system in the left is developed by PERI).

**Figure 3 materials-16-00759-f003:**
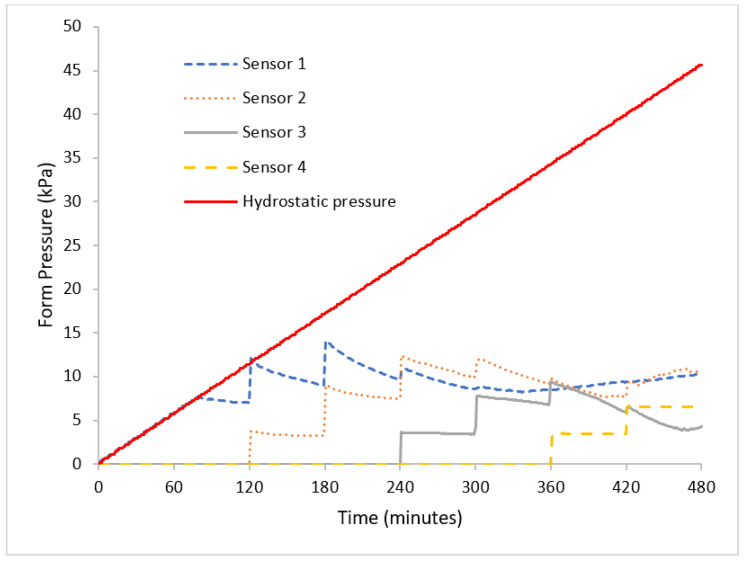
Pressure monitoring over time for 0.25 m/h casting rate.

**Figure 4 materials-16-00759-f004:**
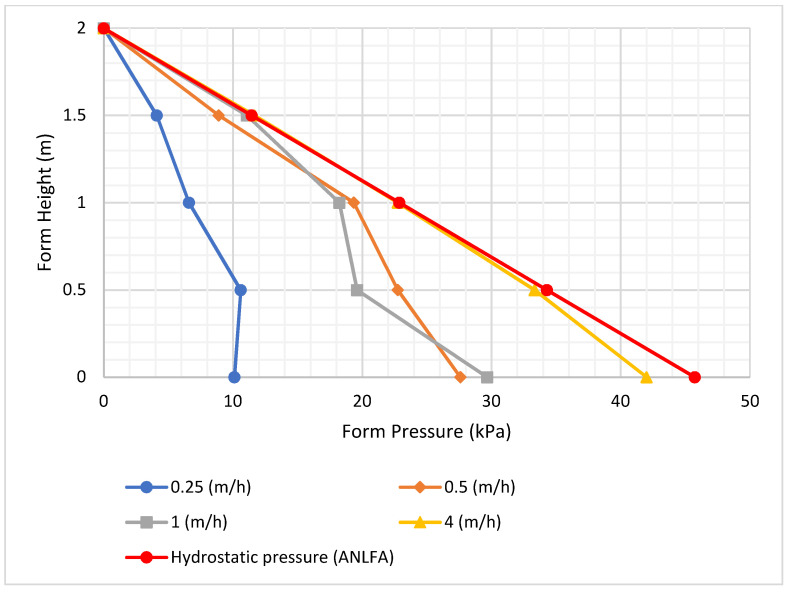
The effect of casting rate on pressure.

**Figure 5 materials-16-00759-f005:**
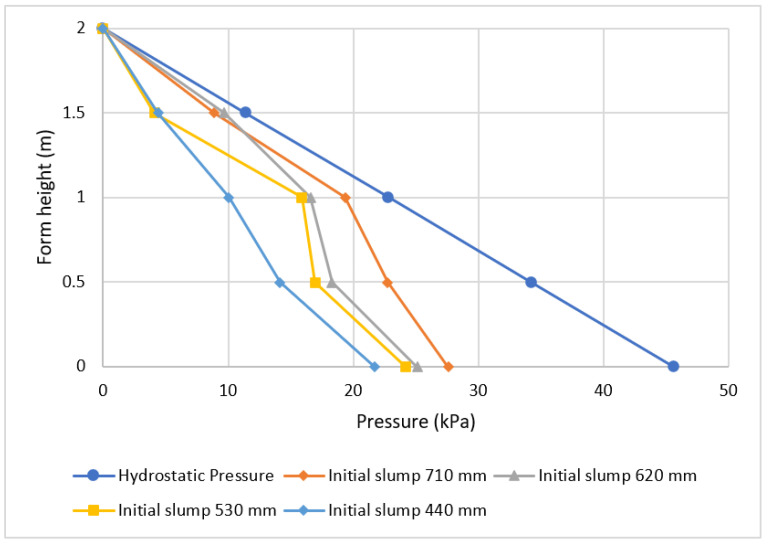
Effect of initial slump flow on pressure.

**Figure 6 materials-16-00759-f006:**
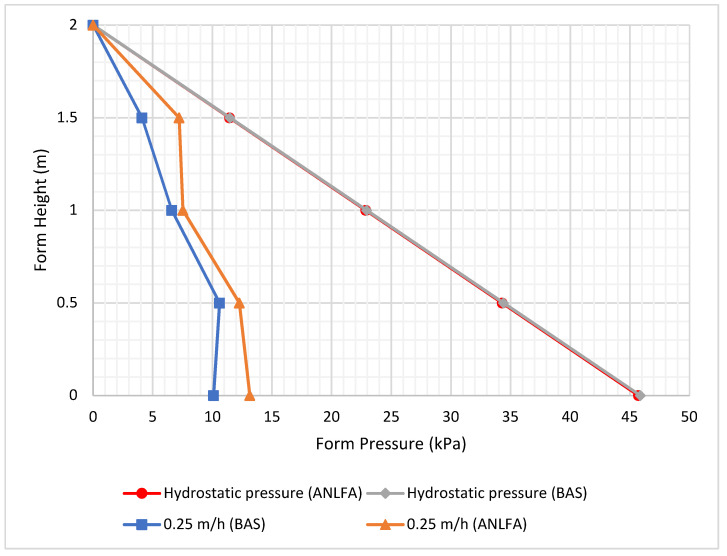
Effect of different types of cement on form pressure.

**Figure 7 materials-16-00759-f007:**
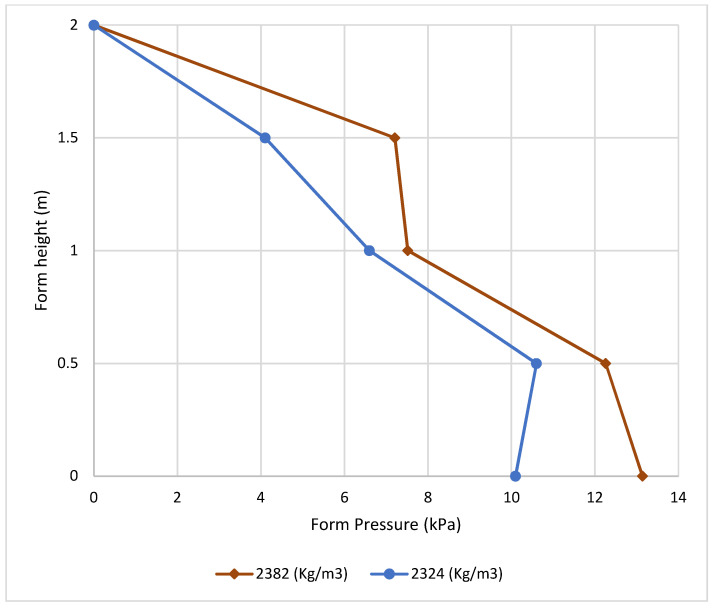
Effect of fresh concrete density on form pressure.

**Figure 8 materials-16-00759-f008:**
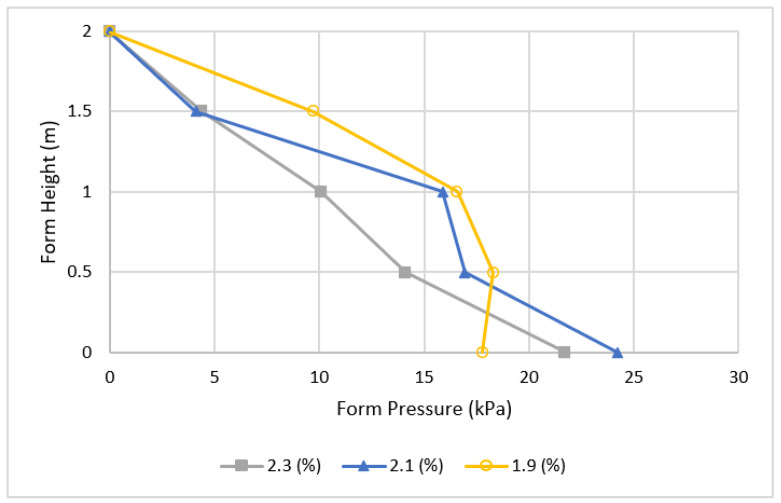
The effect of different air content on pressure.

**Figure 9 materials-16-00759-f009:**
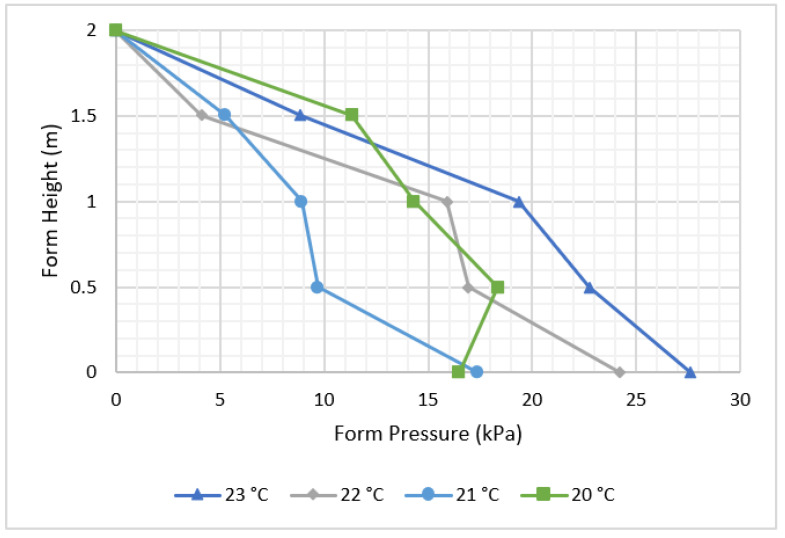
Effect of ambient temperature on form pressure.

**Figure 10 materials-16-00759-f010:**
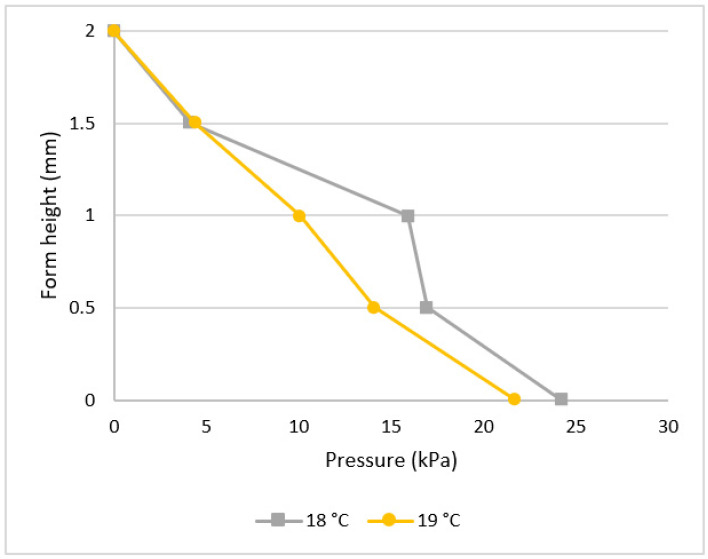
Effect of concrete temperature on form pressure.

**Figure 11 materials-16-00759-f011:**
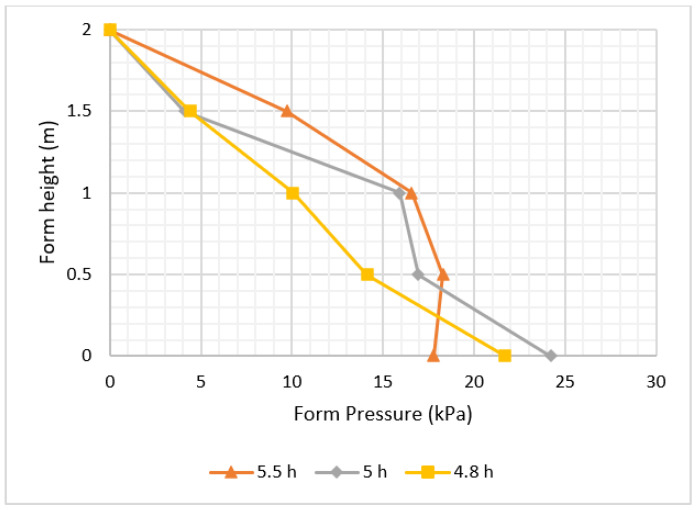
The effect of concrete setting time on pressure.

**Figure 12 materials-16-00759-f012:**
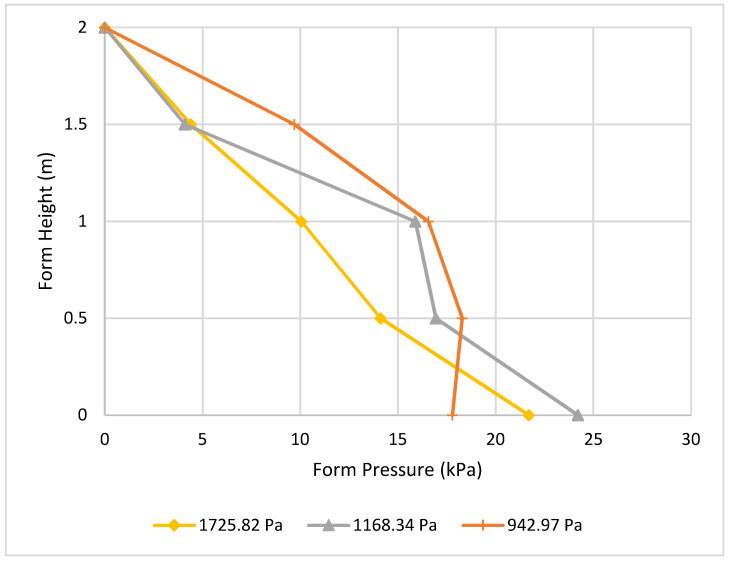
Effect of static yield stress at 15 min on form pressure.

**Table 1 materials-16-00759-t001:** Mix proportions.

Material	Cement: BAS/ANLFA	Filler	Agg (0–8) mm	Agg (8–16) mm	Superplasticizer (MG592)	Water
Kg/m^3^	350	140	978.1	652.08	Varies according to the required slump flow	207

**Table 2 materials-16-00759-t002:** Test plan and parameter variations.

Test No	Targeted Slump Flow (mm)	Casting Rate (m/h)	Cement Type	Variations
1	700–750	0.25	CEM II/A-V 52.5 N (BAS)	Casting rate
2		0.5		
3		1		
4		4		
5	700–750	0.5		Slump flow
6	600–650			
7	500–550			
8	400–450			
9	700–750	0.25	CEM II/A-V 42.5 N (ANLFA)	Cement type, casting rate
10		0.5		
11		4		
12	700–750	0.5		Cement type, slump flow
13	600–650			
14	500–550			

**Table 3 materials-16-00759-t003:** Correlation between input variables and form pressure.

	Density (Kg/m^3^)	Air Content (%)	Air Temperature °C	Concrete Temperature °C	Setting Time (h)	Initial Slump Flow (mm)	Initial T500 Time (s)	Static Yield Stress at 15 min	Height at Pmax	Pmax
Density (Kg/m^3^)	1.00	−0.16	−0.67	0.01	0.59	−0.29	0.60	0.10	0.22	**−0.37**
Air content (%)	−0.16	1.00	0.19	−0.17	−0.51	−0.35	−0.05	0.28	−0.14	0.03
Air temperature °C	−0.67	0.19	1.00	−0.64	−0.34	0.21	−0.27	−0.18	0.11	**0.32**
Concrete temperature °C	0.01	−0.17	−0.64	1.00	0.07	0.16	−0.25	0.03	−0.25	−0.08
Setting time (h)	0.59	−0.51	−0.34	0.07	1.00	0.58	0.72	−0.71	0.27	−0.04
Initial slump flow (mm)	−0.29	−0.35	0.21	0.16	0.58	1.00	0.22	−0.94	0.15	0.15
Initial T500 time (s)	0.60	−0.05	−0.27	−0.25	0.72	0.22	1.00	−0.51	0.34	−0.08
Static yield stress at 15 min	0.10	0.28	−0.18	0.03	−0.71	−0.94	−0.51	1.00	−0.18	−0.19
Form height (m)	0.22	−0.14	0.11	−0.25	0.27	0.15	0.34	−0.18	1.00	**−0.42**
Pmax	**−0.37**	0.03	**0.32**	−0.08	−0.04	0.15	−0.08	−0.19	**−0.42**	1.00

## Data Availability

No available data.

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
