# Peer review of "The Impact of Different Parameters on the Formwork Pressure Exerted by Self-Compacting Concrete"

_materials, 2023, doi:10.3390/ma16020759_

Round 1

Reviewer 1 Report

This manuscript deals with the effect of some parameters, such as casting rate and slump flow, on the formwork pressure of self-compating concrete. The manuscript reports the experimental methodology the results are presented and discussed. This paper is an interesting work with some novel results, which is suitable for Materials. However, the manuscript needs improvement. The following comments should be addressed before the manuscript can be recommended for publication:

-The authors' affiliations should follow the journal guidelines: The PubMed/MEDLINE standard format is used for affiliations: complete address information including city, zip code, state/province, and country.

-The English writing and style should be improved.

-In that abstract and the rest of the manuscript, use the term "T500 test" or "T500 time test" instead of just "T500".

-Please emphasize the novelty of the manuscript. What is new when compared to previous work on the topic?

-In Section 2.1, although the height of the plastic pipe is specified, the height of the concrete column is not. Could you please add this information?

-In Section 2.3, add a Table with all mix constituents' proportions (quantities) for a theoretical mixture with a volume of 1 cubic meter.

-In Section 2.3, add a Table with all the tested cases.

-Did you perform any repeats? Please specify this.

-Sections 3, 4, 5 and 6 should be subsections of Section 2 as they are part of the methodology.

-In Section 7, the quality of the graphs is poor. It looks like Excel plots that were done quickly. Please improve the quality of all the graphs (font size, colours, frame, labels, gridlines etc. Also, homogenize the size and format of all the graphs presented in the results section.

-In Section 7.1, add a formula to calculate hydrostatic pressure, i.e., the red line in Fig. 3.

-The figure at the end of page 5 (the effect of casting rate on the pressure.) should be Fig. 4 instead of Fig. 3. Please fix the numbering of all figures.

-Please combine Fig. 3 and Fig. 4 into a single Figure, Fig.3a and Fig. 3b. Do the same with other figures. This will improve the readability of the paper.

-Fig. 5 (different types of cement) is positioned before the text that mentions the figure. The figure should always appear after the paragraph where it is mentioned. 

-Section 7.2 should be vastly improved. The discussion is very short. I expect at least half a page of text with references. The authors could use their previous review work to discuss their results better.

Reviewer 2 Report

Overall, the study provided some information on the parameters' impact on SCC formwork pressure, which is valuable for the industry. But the quality of the paper needs to be improved before it qualifies for publication. There are a few points the reviewer would like to stress for the revision of the manuscript.

1. The figures may need to be reorganized. There seems to be a weak logic connection in the order of the results. Subtitles may help. And also, somehow, you have two "Figure 5", two "Figure 6", and two "Figure 7". Please be careful and check through the manuscript.

2. The discussions on the results are relatively thin. Please try to discuss the potential contribution of the experiment results, and also, some analysis or discussion on the mechanism would be great. 

3. Table 1 needs more description. Is this p-value? What's the meaning of the data? What's the purpose of the highlighting? Also, a small suggestion here, as a symmetrical table, it can be simplified to eliminate the duplicated data. Please check the format (line space etc.) to make the table looks more concise. 

4. The title and subtitle numbering seem wrong. Please check.

5. Need more information on the test plan. A table with the sample parameters and all expected test outcomes will help. 

6. A thorough grammar and writing check would help to improve the quality of the descriptions.

Round 2

Reviewer 1 Report

The authors have addressed the reviewer's comments. I recommend the paper for publication.

Reviewer 2 Report

The comments have been addressed. The manuscript has been improved.